# Respiratory Health Inequities among Children and Young Adults with Cerebral Palsy in Aotearoa New Zealand: A Data Linkage Study

**DOI:** 10.3390/jcm11236968

**Published:** 2022-11-25

**Authors:** Alexandra Sorhage, Samantha Keenan, Jimmy Chong, Cass Byrnes, Amanda Marie Blackmore, Anna Mackey, Timothy Hill, Dug Yeo Han, Ngaire Susan Stott

**Affiliations:** 1Paediatric Orthopaedics, Starship Children’s Health, Auckland 1023, New Zealand; 2Kidz First Paediatrics, 100 Hospital Road, Auckland 2025, New Zealand; 3Paediatric Rehabilitation Service, Starship Children’s Health, Auckland 1023, New Zealand; 4Paediatric Respiratory Service, Starship Children’s Health, Auckland 1023, New Zealand; 5Department of Paediatrics, University of Auckland, Auckland 1010, New Zealand; 6Telethon Kids Institute, 15 Hospital Ave., Perth 6009, Australia; 7Starship Research and Innovation, Starship Children’s Health, Auckland 1023, New Zealand; 8Department of Surgery, University of Auckland, Auckland 1023, New Zealand

**Keywords:** cerebral palsy, respiratory, hospitalisations, antibiotics, Indigenous, inequities

## Abstract

(1) Background: Respiratory disease is a leading cause of morbidity, mortality, and poor quality of life in children with cerebral palsy (CP). This study describes the prevalence of CP-related respiratory disease and the non-modifiable risk factors for respiratory-related hospital admissions in the Aotearoa New Zealand population. (2) Methods: New Zealand Cerebral Palsy Register (NZCPR) participant data and de-identified data from the National Minimum Dataset and Pharmaceutical Dispensing Collections were linked to identify all respiratory-related hospital admissions and respiratory illness-related antibiotic exposure over 5 years in individuals with CP (0–26 years). (3) Results: Risk factors for respiratory-related hospital admissions included being classified Gross Motor Function Classification System (GMFCS) IV or V compared to GMFCS I [OR = 4.37 (2.90–6.58), *p* < 0.0001; OR = 11.8 (7.69–18.10), *p* < 0.0001, respectively,]; having ≥2 antibiotics dispensed per year [OR = 4.42 (3.01–6.48), *p* < 0.0001]; and being of Māori ethnicity [OR = 1.47 (1.13–1.93), *p* < 0.0047]. Māori experienced health inequities compared to non-Māori, with greater functional disability, and also experienced greater antibiotic dispensing than the general population. (4) Conclusion: Māori children and young adults have a higher risk of respiratory-related illness. Priority should be given to the screening for potentially modifiable risk factors for all children with CP from diagnosis onwards in a way that ensures Māori health equity.

## 1. Introduction

Cerebral palsy (CP) is the most common childhood physical disability in high income countries, with a prevalence between 1.4 and 2.1 cases per 1000 live births [1]. Respiratory-related illness in CP is the most common cause of mortality, morbidity, and poor quality of life, especially for those living with greater disability [≈26% of the CP population is classified as combined Gross Motor Function Classification System (GMFCS) IV and V] [2,3]. Adults with CP have a 14-fold risk of death from respiratory disease compared to the general population [4], and more severely impaired children with CP have a higher frequency of hospital admissions and a longer length of stay related to respiratory illness, with a greater risk of early mortality [5]. The pathway to respiratory disease in children with CP is complex and multifactorial and impacts all levels of disability, with 45% of all people with CP having respiratory symptoms [6].

In Aotearoa New Zealand (AoNZ), the general paediatric population (<15 years of age) experience the highest respiratory hospitalisation rates (comparable only to adults >65 years of age), with hospital admissions for bronchiolitis, asthma, wheeze and viral pneumonia having increased to over 21,000 annually since 2000 [7]. The rates and severity of bronchiectasis are also higher than documented in other developed countries [8,9].

Significant health inequities for Indigenous NZ Māori compared with NZ European are well-recognised [10,11], with Māori children and young people in particular experiencing adverse health and social outcomes as a result of socio-political and economic environmental drivers [12,13]. The Impact of Respiratory Disease in NZ report (2018) stated that ‘All indicators showed inequality in health by ethnic group. Pacific peoples and Māori shared the highest respiratory health burden’ [7]. For respiratory infections, the relative risks are 1.7–18.2 for Māori and Pasifika as compared to NZ European, ranging from 2.1 for pneumonia to 3.9 for bronchiolitis between the most and least deprived areas of socioeconomic deprivation [10]. This indicates that NZ Māori living with CP—especially those classified as GMFCS IV and V—are likely to be more vulnerable to respiratory disease.

CP Registry information and hospital admission data have been recently used in Australia to describe the impact of respiratory disease for people with CP, and three non-modifiable predictive risk factors for future respiratory-related hospital admission were identified: (i) GMFCS V; (ii) ≥1 respiratory-related hospital admission in the previous year; and (iii) ≥2 courses of antibiotics in the previous year [14]. In AoNZ, less is known about the impact of respiratory illness in children with CP, and although hospital admissions have increased over 14 years, the reasons for hospitalisation are not well-documented [15]. Most CP hospitalisations in the 0–24-year age-range have a primary diagnosis of CP (42%), with associated diagnoses including respiratory disease (11%) [16].

The primary aim of this study was to describe the prevalence of respiratory disease and non-modifiable risk factors for hospital admissions for children and young people in AoNZ. In particular, this study aimed to identify any inequities between Indigenous NZ Māori children with CP and other ethnic groups, between the GMFCS levels, and between the different levels of socioeconomic deprivation. The secondary aim of the study was to describe the dispensing of antibiotics for respiratory-related illness in children with CP and to examine the differences between ethnic groups, GMFCS levels, and levels of socioeconomic deprivation.

## 2. Materials and Methods

### 2.1. Study Design

A 5-year (2014–2019) retrospective cohort design study used data linkage for all children and young people diagnosed with CP and registered with the NZCPR who met the following criteria: (i) aged 0–26 years, (ii) residing in AoNZ at the time of data extraction and matched to de-identified data on hospitalisations and antibiotic dispensing for respiratory infections.

### 2.2. Data Sources

The NZCPR is a national register established in 2015 with ethics approval (HDEC 13/NTA/130) in order to collect health data and information relevant to CP (including demographic data and clinical information such as type, topography, and GMFCS). This includes the ability to link to Health New Zealand (previously known as the Ministry of Health) datasets. Data were extracted from the NZCPR and linked to Health New Zealand datasets using the unique electronic National Health Index (NHI) number [17]. The NHI was encrypted, and only de-identified data were used for analysis.

Each hospital is required to code and report hospital admissions (including emergency department admissions >3 h in duration) information to Health New Zealand at monthly intervals. The collective database is called the National Minimum Dataset (also known as Hospital Events data) [18]. Primary and secondary diagnoses at the time of discharge are coded using the International Statistical Classification of Diseases (ICD). The ICD 10th Revision codes included for analysis were J09-J99 as a primary or secondary diagnosis only (excluding J30-J39 for other diseases of upper respiratory tract) for all hospital admissions years 2014–2019.

In AoNZ, antibiotics for systemic use are only available with a prescription, and, at the time of the study, were dispensed free of charge for children aged 13 years or under and incurred a nominal charge for those aged >14 years. Prescriptions dispensed from a community pharmacy seeking government subsidisation are recorded in the Health New Zealand database known as the Pharmaceutical Collection [19]. This does not include antibiotics dispensed in hospital or by a medical practitioner directly. Data for antibiotics dispensed were classified using the Anatomical Therapeutic Chemical (ATC) classification system under agents “for systemic use” defined as “anti-bacterials”. Antibiotics were identified by individual names (e.g., amoxicillin trihydrate). The antibiotics that were more likely to be used in clinical practice to treat respiratory-related infections were identified by one of the authors (Cass Byrnes) who is a Paediatric Respiratory Specialist (Appendix A). A single course was defined as one course if dispensed on separate days, irrespective of the days separating dispensing events. A single course was also defined as one course, even if multiple antibiotics had been dispensed on the same day.

In Health New Zealand datasets, ethnicity is linked to the NHI and is usually collected at the point of the person’s contact with hospital services. Prioritised ethnicity was used for analysis. Prioritisation is a method that assigns people who identify with more than one ethnic group to a single mutually exclusive category based on an established hierarchy, with the aim of giving priority to non-European groups and special priority to Māori and Pacific Island groups [20]. The ethnicity coding used was in line with Statistics New Zealand level 1 categories for national standards reporting [European; NZ Māori; Pasifika (also referred to as Pacific Island Peoples); Asian; and MELAA -Middle Eastern/Latin American/African] [21].

Based on nine Census variables, the NZDep is an area-based measure of socioeconomic deprivation in New Zealand, measuring the level of deprivation for people within a small (10%) geographical area. Each NZDep decile contains 10 deciles, with decile 1 representing the least deprived score and 10 the most deprived [22]. For analysis, the deprivation indices were assigned to quintiles, with quintile 1 representing the 20% least deprived data zone and quintile 5 representing the 20% most deprived zone. Due to the unreliability of the 2018 census data, the NZDep 2013 was used. 

### 2.3. Statistical Analysis

A total of 1286 NZCPR participants met the inclusion criteria. Descriptive statistics were used to summarise the cohort using cross-tabulation and median IQR and range. Chi square and Logistic regression were carried out for binary dependent variables. Multivariate regression was carried out using significant factors from univariate regression where appropriate.

Population data between the 5 yearly censuses were adjusted by linear interpolation to calculate the total population aged under 27 years in each geographical region for 2014–2020 using 2013 and 2018 Census data between ethnic groups (Asian, Māori, Pasifika, European, and MELAA). Since the observed data are the counts of admission and dispensing in each year, the data were analysed using a generalised linear model, modelling the observed counts data as having a Poisson distribution and plotted as a fitted line. A *p* value of < 0.05 was considered to be statistically significant.

Statistical analyses were carried out using SAS 9.4 (SAS Institute Inc., Cary, NC, USA.) and R (R Core Team (2021). R: A language and environment for statistical computing. R Foundation for Statistical Computing, Vienna, Austria [23].

## 3. Results

### 3.1. Cohort Description

Full age, gender, and ethnicity data were available for 1286 participants, with missing data for the GMFCS in 73 (6%) and the measure of socioeconomic deprivation in 3 (0.3%). At the time of NZCPR data extraction, the median age of participants was 14 years; IQR = 10–19, of which 28 (2%) were <5 years; 856 (67%) were aged between 5 and 17 years; and 402 (31%) aged between 18 and 26 years. The Auckland metro region is home to 33% of AoNZ’s population. Of the present cohort, 542 (42%) were living in this region. Demographic and clinical descriptors are detailed in Table 1. There were 734 (57%) male and 552 (43%) female participants, and 410 (34%) participants were classified as GMFCS IV and V. In terms of ethnicity, the cohort was comparable to the AoNZ general population for this age group [23] with 639 (50%) of the cohort being European, followed by Māori 342 (27%), Pasifika 135 (11%), Asian 137 (11%), and MELAA 32 (2%). Overall, 342 (25%) were living in higher areas of deprivation (quintile 5) compared to 254 (20%) in the least deprived areas (quintile 1); with 177 (28%) of European participants living in quintile 1 compared to 8 (25%) of MELAA, 8 (6%) of Pasifika, and 32 (9%) of Māori participants.

Differences in the distribution of GMFCS classification were found for Māori participants when compared to non-Māori participants (Table 2). When participants were grouped by GMFCS IV-V as compared to GMFCS I-III, Māori participants were more likely to be classified as GMFCS IV- or V than non-Māori participants (*p* = 0.0065).

Māori were also significantly over-represented in quintile 5 (*p* < 0.0001) compared to their non-Māori counterparts (Table 2). Similar findings are found for Māori in the general AoNZ population [12].

### 3.2. Respiratory-Related Hospital Admissions

Three hundred and fifty three of the 1286 participants (27%) had 1374 hospital admissions related to respiratory illness, with a median admission rate of 2.0, IQR = 1.0–5.0 over the 5-year period. Of the total admissions, 316/1374 (23%) were following Emergency Department (ED) visits, with no difference found in the proportion of ED presentations between Māori and non-Māori participants (*p* = 0.2031). A significantly higher percentage of participants classified as GMFCS IV- V (201/316, 64%) presented to ED than those classified as GMFCS I–III (102/316, 32%, *p* < 0.0001). A significantly higher percentage of those living in quintiles 4 or 5 (180/301, 57%) presented to ED than those living in quintiles 1 or 2 (81/316, 26%, *p* = 0.0009). Median lengths of stay (LoS), excluding ED admissions, were similar for Māori (Median = 3 days, IQR = 1–6) and non-Māori participants (Median = 3 days, IQR = 2–6).

Following a univariate regression analysis for demographic and clinical variables, participants classified as GMFCS IV or V were found to have higher odds of having a respiratory-related hospital admission than participants classified as GMFCS I (Table 3). Māori participants were more likely to have a respiratory-related hospital admission than non-Māori; participants living in the area of highest deprivation (quintile 5) were 50% more likely to have an admission than those living in low deprivation areas (quintile 1); and the participants who had two or more antibiotics dispensed/year were four times more likely to have a respiratory-related hospital admission (Table 3). Through multivariate analysis within the Māori group, it was determined that the odds of respiratory-related hospital admissions were significantly higher in GMFCS IV- V than GMFCS I- III [OR = 5.55 (2.66–11.6) *p* < 0.0001], whereas there was no association with the level of deprivation (quintile 5 compared with quintile 1) [OR = 0.92 (0.36–2.33), *p* = 0.8505]. 

Population data for the age group and timeframe of interest showed that European participants experienced significantly lower rates of admission per 100,000 people in overall years than their Māori counterparts (*p* < 0.001). MELAA participants had a much smaller representation in the cohort but the highest overall admissions of 1.8× (84%) higher than the rate of NZ Māori (*p* < 0.0001) (Figure 1). Hospital admission rates for children and young adults with CP remained relatively steady for European and Asian groups over time but declined significantly for Pasifika (*p* = 0.0003). An apparent decline was found not to be significant for MEELA (*p* = 0.571). NZ Māori were the only group who experienced a trend to higher admission rates over this time. 

### 3.3. Antibiotic Dispensing for Respiratory-Related Illness

Overall, a total of 9647 episodes of antibiotics were dispensed for 1170/1286 (91%) of the cohort with a median of 5 episodes, IQR = 3–11 over the 5-year timeframe, with no differences found between GMFCS, ethnicity, or quintiles. Most individuals had none or only one episode per year, but 151/1286 (12%) had ≥2 episodes of antibiotics/year dispensed over the timeframe. No significant differences were found amongst ethnicities (including Māori and non-Māori) or amongst quintiles for ≥2 episodes of antibiotic dispensing per year; however, univariate analysis found that children classified as GMFCS V were almost three times more likely to have ≥2 episodes of antibiotics dispensed per year than those classified as GMFCS I (Table 4). 

Using population data for the age group and timeframe of interest, overall rates of total antibiotic dispensing for children and young adults with CP decreased over time for all ethnic groups. When compared to Māori, all other ethnic groups had significantly lower rates of antibiotic dispensing, with Asian participants having the lowest rate (−53%) followed by European (−41%), MELAA (−26%), and Pasifika (−17%) (*p* < 0.0001) (Figure 2).

## 4. Discussion

This study has shown that almost all children and young adults with CP had antibiotics dispensed for respiratory-related illness in the community over the study time frame, with high rates of ED presentations and hospital admissions for respiratory illness. One in ten children and young adults with CP had >2 courses of antibiotics per year, with children classified as GMFCS V being almost three times more likely to have ≥2 antibiotics dispensed per year than those classified as GMFCS I. One in five children and young adults with CP experienced a respiratory-related hospital admission, with individual risk factors including being classified as GMFCS IV or V compared to GMFCS I; having ≥2 episodes of antibiotics dispensed per year; living in the most deprived quintile (Q5) compared to the least deprived Q1; and being of NZ Māori ethnicity.

There are few international studies of admission rates with respiratory illness in children with CP. However, a comparable Australian study reported that 11% of Western Australian participants with CP experienced a respiratory-related admission over a 3-year timeframe with GMFCS V experiencing admission rates 23 times higher than GMFCS I [14]. Direct comparison of the data shows that in AoNZ, 27% of our participants had 1374 hospital admissions, an average of 3.6 per person over 5 years. In Western Australia, 16% of participants had 332 respiratory hospital admissions, an average of 4.3 per person over 5 years [24]. These data paint a picture of a high overall risk of respiratory-related hospital admissions in children and young adults with CP, a risk which is much higher than the reported risk in the general population for admissions with other respiratory illnesses such as RSV (reported to range from 1.3 to 10.5 hospitalizations per 1000 in the age group 1–2 years) [25].

In the current study, Indigenous NZ Māori children and young adults with CP experienced a higher incidence of respiratory-related hospital admissions over time when compared to non-Māori participants and were more likely to function at GMFCS levels IV or V. Individuals with CP and GMFCS levels IV or V often have associated co-morbidities, such as oropharyngeal motor dysfunction and recurrent seizures [26,27]. Overall, oropharyngeal motor dysfunction is common in people with CP [25] and results in sequential direct aspiration leading to aspiration pneumonia, acute respiratory infections, and chronic airway inflammation [28]. Similarly, seizures can lead to temporary altered muscle tone and reduced consciousness, which may increase the risk of aspiration, leading to aspiration pneumonia. Not surprisingly then, oropharyngeal motor dysfunction and seizures are major risk factors for an increased risk of hospitalization with respiratory-related illness in children with CP [29]. The association between GMFCS levels and potentially modifiable medical risk factors could explain why NZ Māori, who had greater physical disability in this study, also experienced a higher incidence of respiratory-related hospital admissions over time. Further research should help determine the prevalence of oropharyngeal motor dysfunction and seizure disorder in this group and whether the optimisation of medical management could help to reduce their risk of hospitalisation with respiratory disease.

Trend data for the CP population showed that over the 5 years of the study, the rate of admission to hospital with respiratory illnesses either remained stable or dropped for all ethnic groups. From the trend data, NZ Māori had the second highest rate of hospital admission, second only to MELAA participants. Surprisingly, being of MELAA ethnicity did not reach significance as a risk factor in the analysis of individual risk factors, perhaps due to the very small numbers in this group (32 children). Unlike most countries, AoNZ’s social policy does not bar refugees or asylum seekers on the grounds of medical conditions or disabilities and allocates over half of its international allocation to MELAA regions [30]. It is unknown if this small group of MELAA participants were predominantly an immigrant population, with health outcomes reflecting prior lack of access to health care in their originating countries, or whether they were born in AoNZ and self-identified as MELAA.

This is the first study to use a pharmaceutical dispensing dataset to identify antibiotic use to treat respiratory-related illness in CP. Antibiotics were widely used, with those classified as GMFCS V having significantly higher chances of receiving ≥2 courses/year dispensed. NZ Māori children and young adults with CP consistently had the highest rates of antibiotic episodes per 100,000 of population compared to other ethnicity groups over the 5 years. Hobbs, et al., also reported that Māori children received more antibiotic courses than European children and young adults and suggested that much of this prescription was for self-limiting viral respiratory infections [31]. This suggests that changes in antibiotic prescribing practice for NZ Māori children and young adults may still lag behind changes in prescribing for other ethnic groups.

### Limitations

The NZCPR is not a population-based register and is not fully ascertained, but it is representative of 60% of the estimated CP population in AoNZ; ascertainment strategies have been biased towards tertiary care/hospital ascertainment, which could account for a slightly higher proportion of GMCFS IV and V when compared to the Australian Cerebral Palsy Register (ACPR GMFCS IV-V 26% compared to NZCPR GMFCS IV-V 34%) [2].

Hospital admissions data do not include community or primary care presentations for respiratory illness, which would further add to the understanding of respiratory illness in the community. We have used antibiotic prescribing data as a proxy for respiratory infection and have followed guidelines to identify those antibiotics commonly used for respiratory infection. Consistency and regional differences in coding (in particular, for ED admissions) have been questioned in the past; however, processes have been universally implemented to significantly improve reliability and it is less likely to be of concern for our timeframe of interest [32,33]. The Pharmaceutical Collection does not account for hospital or GP dispensing, although the latter, especially for antibiotics, would be rare. Moreover, dispensing medication does not always mean it is taken, or taken as prescribed, and although a financial incentive is in place, not all pharmacies need to partake in contributing data. However, participation is thought to be high. The methodology used for defining an episode could overestimate actual antibiotic dispensing, as some episodes may have been an extension of the preceding course; however, it is thought that, taking into consideration the limitations, antibiotic exposure is likely to be underreported [31]. Therefore, the high rates for antibiotic dispensing found in this study are likely to be, if anything, an underestimate of the true levels of antibiotic dispensing and usage. The antibiotics included for analysis may not be exclusively used for respiratory illness. The Mortality Collection dataset was not used to identify mortality with an underlying cause of respiratory illness, as data were available until only 2017. We did not collect respiratory viral panel results or sputum cultures from hospital admissions as these results are not available through dataset linkage, but recognise that those with specific pathogens, such as Pseudomonas aeruginosa, may have more severe and/or chronic respiratory disease [34]. We also did not look at coding for aspiration, an issue that has been associated with recurrent pneumonia in children generally [35] and with cerebral palsy in particular [36]. It is also associated with more significant pathogenic infection [34], although this is likely to align along the GMFCS levels of disability.

## 5. Conclusions and Future Directions

While some factors may not be modifiable, to improve morbidity and mortality, modifiable risk factors require management [37] with the aim of moving from a reactive care model to a surveillance, preventative, whānau-centred (family-centred), and child-centred model. The second phase of the project is to test the feasibility of the CP Respiratory Checklist (developed in Western Australia for health professionals and families) in AoNZ for use in clinical practice as the first step in the shift towards the early identification of modifiable risk factors for respiratory illness [38]. Identifying inequities assists in focusing resources that promote equity in the early years of childhood [39]. All health care professionals caring for children with CP can make positive contributions to eliminating health inequities through the re-orientation of systemic and health determinants and the adoption of equity-focused practices [40].

## Figures and Tables

**Figure 1 jcm-11-06968-f001:**
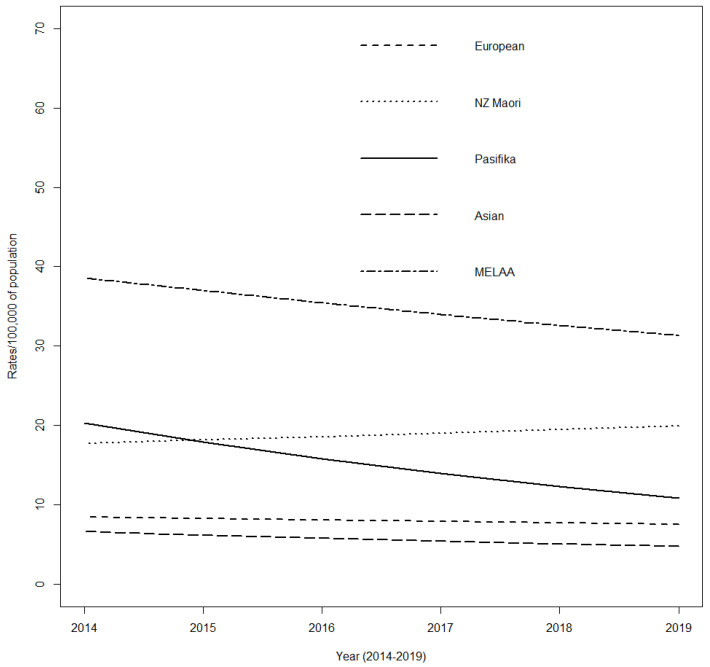
Incidence rates of total hospital admissions for children and young adults with cerebral palsy by ethnicity, Aotearoa New Zealand; 2014–2019.

**Figure 2 jcm-11-06968-f002:**
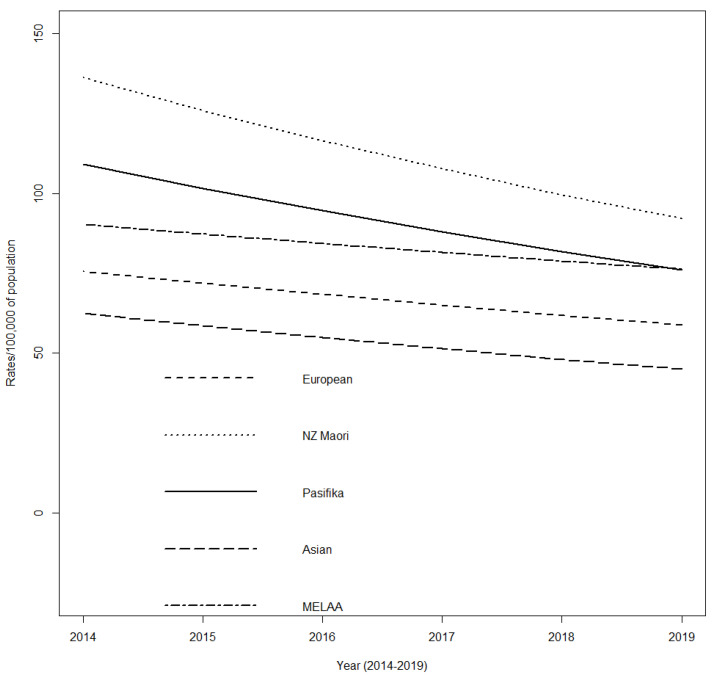
Incidence rates of total antibiotic dispensing for children and young adults with cerebral palsy by ethnicity, Aotearoa New Zealand; 2014–2019.

**Table 1 jcm-11-06968-t001:** Demographics of the Study Participants, including median age and gender distribution, GMFCS, and NZDep quintile distribution by ethnic group.

Cohort Demographic and Clinical Descriptors						
	European	NZ Māori ^1^	Pasifika	Asian	MELAA ^2^	Total
Age (*n* = 1286) Median (range)	15 (2–26)	14 (3–26)	15 (4–26)	13 (2–26)	14 (5–26)	14 (2–26)
Gender (*n* = 1286) M: F	379: 260	186: 156	76: 59	73: 64	20: 12	734:552
GMFCS ^3^ levels (*n* = 1213) I: II: III: IV: V	204: 157: 72: 89: 72	97: 65: 30: 64: 64	30: 31: 7: 35: 30	44: 37: 9: 29: 14	5: 11: 2: 7: 6	381:301:121: 224:186
Unknown GMFCS ^2^ (*n* = 73)	45	22	2	4	0	
NZDep ^4^ Quintiles (*n* = 1283) 1: 2: 3: 4: 5	177: 139: 131: 109: 82	32: 32: 43: 92: 143	8: 11: 13: 39: 63	28: 34: 25: 22: 27	8: 2: 4: 9: 9	254: 218: 216: 271: 324

^1^ Indigenous population in Aotearoa New Zealand, ^2^ Middle Eastern Latin American African, ^3^ Gross Motor Function Classification System. ^4^. NZDep is an area-based measure of socioeconomic deprivation in New Zealand

**Table 2 jcm-11-06968-t002:** Comparisons of Māori and non-Māori children and young adults with CP by age, gender, GMFCS, and NZDep quintiles.

Cohort Descriptors for Māori and Non-Māori			
	NZ Māori ^1^ (N = 320)	Non-Māori (N = 943)	*p*
Age Median (range)	14 (3–26)	15 (2–26)	
Gender (*n* = 1286) M: F	186: 156	548: 395	0.2329
GMFCS ^2^ levels (*n* = 1213) I: II: III: IV: V	97: 65: 30: 64: 64	283: 236: 91: 160: 122	0.0319
NZDep ^3^ Quintiles (*n* = 1286) 1: 2: 3: 4: 5	32: 32: 43: 92: 143	221: 186: 173: 179: 182	<0.0001

^1^ Indigenous population in Aotearoa New Zealand. ^2^ Gross Motor Function Classification System. ^3^ NZDep is an area-based measure of socioeconomic deprivation in New Zealand.

**Table 3 jcm-11-06968-t003:** Univariate regression analysis and odds ratio analysis for respiratory-related admissions for children and young adults with CP for gender, GMFCS, ethnicity, NZDep quintile variables, and if ≥2 antibiotics dispensed per year.

	Respiratory-Related Admissions (*n* = 353)		*p*
	N (%)	Odds Ratio (95% CI)	
Gender (vs. male)			
Male	201 (57)	1.00	
Female	152 (43)	1.01 (0.79–1.29)	0.9518
GMFCS ^1^ Level (vs. Level I)			
I	46 (14)	1.00	
II	70 (21)	2.21 (1.47–3.32)	<0.0001
III	17 (5)	1.19 (0.65–2.17)	
IV	84 (25)	4.37 (2.90–6.58)	
V	115 (35)	11.8 (7.69–18.1)	
GMFCS ^1^ Group (vs. Level I/II/III)			
I/III/III	133 (40)	1.00	
IV/V	199 (60)	4.75 (3.63–6.22)	<0.0001
Ethnicity (vs. NZ Māori ^2^)			
NZ Māori	114 (32)	1.00	
European	152 (43)	0.62 (0.47–0.83)	0.0004
Pasifika	50 (14)	1.18 (0.78–1.78)	
Asian	27 (8)	0.49 (0.31–0.79)	
MELAA ^3^	10 (3)	0.91 (0.42–1.98)	
NZ Māori (vs. non-Māori)			
Non-Māori	239 (68)	1.00	
NZ Māori	114 (32)	1.47 (1.13–1.93)	0.0046
NZDep ^4^ Quintiles (vs. Q1)			
Q1 (least deprived)	55 (16)	1.00	
Q2	62 (18)	1.44 (0.95–2.19)	0.2506
Q3	61 (17)	1.42 (0.94–2.17)	
Q4	79 (22)	1.49 (1.00–2.21	
Q5 (most deprived)	95 (27)	1.50 (1.02–2.20)	
Antibiotic Dispensing			
< 2 Episodes/year	34 (11)	1.00	
≥2 Episodes/year	314 (36)	4.42 (3.01–6.28)	<0.0001

^1^ Gross Motor Function Classification System. ^2^ Indigenous population in Aotearoa New Zealand. ^3^ Middle Eastern Latin American African. ^4^ NZDep is an area-based measure of socioeconomic deprivation in New Zealand. Q = Quintile.

**Table 4 jcm-11-06968-t004:** Univariate analysis and odds ratio analysis for ≥2 antibiotic episodes per year for gender, GMFCS, ethnicity, and NZDep quintile.

	≥2 Antibiotic Episodes/Year (*n* = 151) OR		*p*
	N (%)	Odds Ratio (95% CI)	
Gender (vs. male)			
Male	94 (32)	1.00	
Female	57 (38)	0.79 (0.56–1.13)	0.1926
GMFCS ^1^ Level (vs. Level I)			
I	60 (40)	1.00	
II	30 (20)	1.75 (1.09–2.81)	0.0053
III	15 (10)	1.37 (0.74–2.53)	
IV	21 (14)	1.98 (1.17–3.37)	
V	13 (7)	2.78 (1.48–5.22)	
GMFCS ^1^ Group (vs. Level I/II/III)			
I/III/III	105 (76)	1.00	
IV/V	34 (25)	1.80 (1.20–2.71)	0.0047
Ethnicity (vs. NZ Māori ^2^)			
NZ Māori	36 (24)	1.00	
European	85 (56)	1.42 (0.94–2.15)	0.0862
Pasifika	8 (5)	0.55 (0.25–1.21)	
Asian	18 (12)	1.43 (0.78–2.62)	
MELAA ^3^	4 (3)	1.41 (0.46–4.30)	
NZ Māori (vs. non-Māori)			
Non-Māori	115 (76)	1.00	
NZ Māori	36 (24)	0.78 (0.53–1.17)	0.2291
NZDep ^4^ Quintiles (vs. Q1)			
Q1 (least deprived)	33 (22)	1.00	
Q2	21 (14)	0.68 (0.38–1.22)	0.7604
Q3	27 (18)	0.92 (0.53–1.59)	
Q4	32 (21)	0.83 (0.49–1.39)	
Q5 (most deprived)	38 (25)	0.82 (0.50–1.36)	

^1^ Gross Motor Function Classification System; ^2^ Indigenous population in Aotearoa New Zealand; ^3^ Middle Eastern Latin American African; ^4^ NZDep is an area-based measure of socioeconomic deprivation in New Zealand. Q = Quintile.

## Data Availability

In keeping with the Data Management Protocol as part of the Ethics approval, all data is stored securely within the NZCPR folder held within the Health New Zealand IT system. To access de-identified data, please email your request to nzcpregister@adhb.govt.nz.

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
