# Peer review of "Respiratory Health Inequities among Children and Young Adults with Cerebral Palsy in Aotearoa New Zealand: A Data Linkage Study"

_jcm, 2022, doi:10.3390/jcm11236968_

Round 1

Reviewer 1 Report

Thank you for inviting me to review "Respiratory Health Inequities Among Children and Young Adults with Cerebral Palsy in Aotearoa New Zealand: A Data Linkage Study".

The manuscript presents an interesting study that provides evidence for common sense concerns about health risk factors in populations with chronic diseases. These findings are useful both for the specific population on the study and for other populations in similar conditions, to whom, in thesis, they could be extrapolated.

The issues on this manuscript are mostly about presentation of data, rather than methodology and data themselves.

Through the manuscript it is not easy to understand when the results concern either the general population or the subpopulation with cerebral palsy only.

Keywords are insufficient and could be both more specific and explanatory.

When interpreting their findings and discussing them, Authors first assume quite affirmatively the indicators on the odds for receiving antibiotic therapy (lines 247-249), but they later underline those are based on untested assumptions (4.1. Limitations). A more prudent approach is recommended.

As Authors decided to analyse original decile NZDep data transformed to quintiles, when interpreting the results of the analysis, it is not correct to extrapolate to deciles the results obtained with quintiles.

The statement on the influence of feeding difficulties (lines 274-278) needs to be rephrased and better sustained.

Tables have short titles only; a short presentation/interpretation of the main information on each table would be clarifying and increase their interest.

Tables miss the insertion of the meaning of the acronyms. The mention of “Quintiles” should be preceded by “NZDep”.

Figures have some names on the Ethnicity classes different from those used in the text.

The manuscript requires review of grammatic (line 141 – “were”, nor “was”) and spelling (line 145 – “easch”). Some terms are written either with its initial both on upper and lower case (“Quintile” vs. “quintile”), others are used with different spelling (“Pacific” [line 146] vs. “Pasifika” [line 130]).

Line 43 - GMFCS stands for “Gross Motor Function Classification System”, not “Functional”.

Line 44-45 – a time or age frame for dying should be stated.

Line 158 – “NZDep13”?

Lines 169 to 171 – use always the same sequence of absolute and relative (%) numbers.

Line 178 – insert paragraph; different issues are presented.

Lines 270-271 – correct the position of “as” in the sentence.

All these issues are easily corrected.

Author Response

The issues on this manuscript are mostly about presentation of data, rather than methodology and data themselves.

Response: Thank you for the opportunity to revise the presentation of our data, changes have been made to the manuscript that clarify the presentation of results and discussion and address the points below:

Through the manuscript it is not easy to understand when the results concern either the general population or the subpopulation with cerebral palsy only.

Response: Changes have been made to the text and figure headings to clarify the results and incidence rates are related to the CP sub-population (children and young adults with CP)

Keywords are insufficient and could be both more specific and explanatory.

Response: key words have been expanded to be more specific and explanatory

When interpreting their findings and discussing them, Authors first assume quite affirmatively the indicators on the odds for receiving antibiotic therapy (lines 247-249), but they later underline those are based on untested assumptions (4.1. Limitations). A more prudent approach is recommended.

Response: Changes above made in recognition that both respiratory related admissions and antibiotic dispensing as markers of respiratory disease in people with CP. Despite limitations in antibiotic dispensing data, a previous study using this dataset for all children under 5years of age in AoNZ felt that overall, antibiotic dispending is underestimated and therefore the rates could be even higher in CP that what we have reported. Focussing on the high risk group of those who had = or >2 episodes, this still identified significant differences for the group in GMFCS V, as compared with GMFCS I that have been highlighted (leading into second sentence in the discussion).

As Authors decided to analyse original decile NZDep data transformed to quintiles, when interpreting the results of the analysis, it is not correct to extrapolate to deciles the results obtained with quintiles.

Response: Reference to decile has been removed from the discussion and referred to as quintile (Line 280)”… with socio-economic quintile being non-contributory.”

The statement on the influence of feeding difficulties (lines 274-278) needs to be rephrased and better sustained.

Response: This discussion point has been extended and elaborated on, with additional references added to support statements: Overall, oropharyngeal motor dysfunction is common in people with CP [26] and results in sequential direct aspiration leading to aspiration pneumonia, acute respiratory infections and chronic airway inflammation [27].  Similarly, seizures can lead to temporary altered muscle tone and reduced consciousness, which may increase the risk of aspiration, leading to aspiration pneumonia. Not surprisingly then, oropharyngeal motor dysfunction and seizures are major risk factors for increased risk of hospitalization with respiratory related illness in children with CP [24]. The association between GMFCS levels and potentially modifiable medical risk factors could explain why Indigenous NZ Māori, who had greater physical disability in this study, also experienced a higher incidence of respiratory-related hospital admissions over time. Further research should help determine the prevalence of oropharyngeal motor dysfunction and seizure disorder in this group and whether optimisation of medical management could help reduce their risk of hospitalisation with respiratory disease.

Tables have short titles only; a short presentation/interpretation of the main information on each table would be clarifying and increase their interest.

Response: Table titles 1-3 have been expanded. We have also expanded titles of the figures 1-2 to provide clarity that the admissions rates and antibiotic dispensing rates are for the CP sub population for ethnicity

Tables miss the insertion of the meaning of the acronyms. The mention of “Quintiles” should be preceded by “NZDep”.

Response: Tables 1-3 have been amended to reflect heading “NZDep Quintile” both in table content and headings

Figures have some names on the Ethnicity classes different from those used in the text.

Response: Pacific Island has now been changed to Pasifika in both figure 1 and figure 2 to correlate to text

The manuscript requires review of grammatic (line 141 – “were”, nor “was”) and spelling (line 145 – “easch”). Some terms are written either with its initial both on upper and lower case (“Quintile” vs. “quintile”), others are used with different spelling (“Pacific” [line 146] vs. “Pasifika” [line 130]).

Response: Grammar and spelling throughput have been corrected. All capitalisation of Quintiles in the text has been removed but retained as a capital letter in the table content. The use of Pasifika is now consistent in the text, however we have made reference to the fact that the term can be used interchangeably with Pacific Island Peoples in line 130, this reflects the interchangeability of the two terms used in AoNZ

Line 43 - GMFCS stands for “Gross Motor Function Classification System”, not “Functional”.

Response: Functional has now been changed to Function. This was also identified as incorrect in the abstract and corrected

Line 44-45 – a time or age frame for dying should be stated.

Response: The data are taken from a longitudinal study of adults beginning at a median age of 29 years and following them for a variable number of years, so past 60 years of age. So it is not possible to set a time frame for age of death without being misleading.

Line 158 – “NZDep13”?

Response: NZDep13 has been removed from this sentence; we feel there is a sufficient description of the measure of socioeconomic measurement in 2.2.5 to warrant not using it in the results text.

Lines 169 to 171 – use always the same sequence of absolute and relative (%) numbers.

Response: This has now been changed to read 8 (6%) of Pasifika, in keeping with the sequence

Line 178 – insert paragraph; different issues are presented.

Response: Paragraph has now been inserted to separate GMFCS and socioeconomic themes

Lines 270-271 – correct the position of “as” in the sentence.

Reviewer 2 Report

This paper was well written, with clear aims around an important issue, methodology and outcomes. I particularly appreciated the information provided about the AoNZ context, for example with MELAA refugee policy.

I was a little confused around the relationship between Maori status and antibiotic dispensing. The data in table 4 shows little difference between Maori/non-Maori status regarding at least 2 episodes/yr (NZCPR data-linkage), yet the data in figure 2 shows significantly higher dispensing rates/100,000 persons/yr for those of Maori background (population data). It would be helpful to add a little more description or discussion.

Author Response

Thank you for the opportunity to revise the presentation of our data, changes have been made to the manuscript that clarify the presentation of results and discussion and address the point below:

I was a little confused around the relationship between Maori status and antibiotic dispensing. The data in table 4 shows little difference between Maori/non-Maori status regarding at least 2 episodes/yr (NZCPR data-linkage), yet the data in figure 2 shows significantly higher dispensing rates/100,000 persons/yr for those of Maori background (population data). It would be helpful to add a little more description or discussion.

Response: Changes have been made to the text in both the results and discussion sections and figure headings to clarify the results and incidence rates are related to the CP sub-population (children and young adults with CP) for ethnicity. The results refer to incidence rates of total antibiotic dispensing/ hospital admissions for children and young adults with cerebral palsy by ethnicity.